# ICAM-1-suPAR-CD11b Axis Is a Novel Therapeutic Target for Metastatic Triple-Negative Breast Cancer

**DOI:** 10.3390/cancers15102734

**Published:** 2023-05-12

**Authors:** Dong Li, Hami Hemati, Younhee Park, Rokana Taftaf, Youbin Zhang, Jinpeng Liu, Massimo Cristofanilli, Xia Liu

**Affiliations:** 1Department of Toxicology and Cancer Biology, College of Medicine, University of Kentucky, Lexington, KY 40536, USA; dong.li@uky.edu (D.L.);; 2Department of Medicine, Hematology/Oncology Division, Feinberg School of Medicine, Northwestern University, Chicago, IL 60611, USA; 3Markey Cancer Center, University of Kentucky, Lexington, KY 40536, USA; 4Robert H. Lurie Comprehensive Cancer Center, Feinberg School of Medicine, Northwestern University, Chicago, IL 606011, USA; 5Department of Medicine, Division of Hematology and Medical Oncology, Weill Cornell Medicine, New York City, NY 10065, USA

**Keywords:** TNBC, metastasis, neutrophil, soluble uPAR, ICAM-1, CD11b, CTC–neutrophil cluster

## Abstract

**Simple Summary:**

The binding of neutrophils with circulating tumor cells (CTCs) enhances the metastatic ability of CTCs. However, the mechanism by which neutrophils bind to CTCs remains elusive. In this study, we found that intercellular adhesion molecule-1 (ICAM-1) on triple-negative breast cancer (TNBC) cells and CD11b on neutrophils mediate tumor cell–neutrophil binding. Consequently, CD11b deficiency inhibits tumor cell–neutrophil binding and metastasis. Moreover, we found that ICAM-1 in TNBC cells promotes tumor cells to secrete soluble urokinase-type plasminogen activator receptor (suPAR), which functions as a chemoattractant for neutrophils. Knockdown of uPAR in TNBC cells reduced lung-infiltrating neutrophils and lung metastasis. Our findings suggest that blocking the ICAM-1-suPAR-CD11b axis might be a novel therapeutic strategy to inhibit TNBC metastasis.

**Abstract:**

Accumulating evidence demonstrates that circulating tumor cell (CTC) clusters have higher metastatic ability than single CTCs and negatively correlate with cancer patient outcomes. Along with homotypic CTC clusters, heterotypic CTC clusters (such as neutrophil–CTC clusters), which have been identified in both cancer mouse models and cancer patients, lead to more efficient metastasis formation and worse patient outcomes. However, the mechanism by which neutrophils bind to CTCs remains elusive. In this study, we found that intercellular adhesion molecule-1 (ICAM-1) on triple-negative breast cancer (TNBC) cells and CD11b on neutrophils mediate tumor cell–neutrophil binding. Consequently, CD11b deficiency inhibited tumor cell–neutrophil binding and TNBC metastasis. Furthermore, CD11b mediated hydrogen peroxide (H_2_O_2_) production from neutrophils. Moreover, we found that ICAM-1 in TNBC cells promotes tumor cells to secrete suPAR, which functions as a chemoattractant for neutrophils. Knockdown of uPAR in ICAM-1^+^ TNBC cells reduced lung-infiltrating neutrophils and lung metastasis. Bioinformatics analysis confirmed that uPAR is highly expressed in TNBCs, which positively correlates with higher neutrophil infiltration and negatively correlates with breast cancer patient survival. Collectively, our findings provide new insight into how neutrophils bind to CTC to facilitate metastasis and discover a novel potential therapeutic strategy by blocking the ICAM-1-suPAR-CD11b axis to inhibit TNBC metastasis.

## 1. Introduction

Breast cancer is the most common cancer in women, and metastasis causes 90% of breast cancer deaths. Among all types of breast cancers, triple-negative breast cancer (TNBC, also known as basal-like breast cancer), tends to be more aggressive and metastatic, and often has a worse prognosis than other subtypes [1,2,3,4,5,6]. Since TNBC lacks estrogen receptor, progesterone receptor, and human epidermal growth factor receptor 2 (HER2), patients do not respond to treatments that target hormone receptors or HER2. Few targeted therapies are available to treat TNBC. Hence, there is an urgent need to better understand the mechanisms underlying TNBC metastasis and to discover novel therapeutic targets for this lethal disease.

To metastasize, cancer cells must enter the circulatory system. These circulating tumor cells (CTCs) are considered the seeds of metastasis and have been used as a liquid biomarker to predict patient survival and treatment response [7,8,9]. Recently, our studies and others have demonstrated that homotypic CTC clusters possess a higher metastatic potential than single CTCs, and patients with homotypic CTC clusters have worse outcomes than patients without them [10,11,12,13]. In addition to homotypic CTC clusters, heterotypic CTC clusters (such as neutrophil–CTC clusters) have also been identified in both mouse tumor models and cancer patients [14,15]. Importantly, CTC-associated neutrophils provide a proliferative advantage for CTCs, leading to more efficient metastasis formation compared to homotypic CTC clusters, and worse patient outcomes [14,16]. However, the underlying mechanism by which neutrophils bind to CTC is not fully understood.

Urokinase-type plasminogen activator receptor (uPAR), encoded by the PLAUR gene, belongs to the urokinase plasminogen activator system [17]. uPAR is highly expressed in a variety of tumor tissues and plays important roles in tumor invasion and metastasis [17,18,19,20]. One of the unique features of uPAR is that the entire protein can be shed from the cell surface by a specific phospholipase C [21]. The soluble form of uPAR (suPAR) is present in the blood of cancer patients, and the increased suPAR level is associated with poor patient prognosis in several types of cancers [22,23,24,25,26]. However, whether and how suPAR is involved in TNBC metastasis is yet to be determined.

CD11b is a protein subunit that, together with CD18, forms macrophage-1 antigen (Mac-1 or integrin α_M_β_2_) [27,28]. CD11b is highly expressed on the surface of neutrophils and regulates neutrophil adhesion and migration [28]. Intercellular adhesion molecule-1 (ICAM-1), which is typically expressed on endothelial and immune cells, is the primary endothelial adhesive ligand for CD11b during trans-endothelial migration of neutrophils [29]. Recently, by comparing primary tumor cells with cells metastasized into the lungs using single-cell RNA sequencing, we identified that ICAM-1 in TNBC cells is a new driver for TNBC metastasis [30]. ICAM-1 expression was dramatically increased in the lung metastases, and mediated CTC homotypic cluster formation [30]. Surprisingly, we frequently found ICAM-1^+^ CTCs are also associated with neutrophils. Since CTC-associated neutrophils can enhance the metastatic ability of CTCs [14,16], it is important to further understand how ICAM-1^+^ CTCs are bound with neutrophils, which could lead to the development of novel strategies to inhibit TNBC metastasis.

Here, we report that ICAM-1 on TNBC cells and CD11b on neutrophils mediate tumor cell–neutrophil binding, providing new insight into the mechanism of CTC–neutrophil cluster formation. In the meantime, CD11b on the neutrophils enhances reactive oxygen species (ROS)/hydrogen peroxide (H_2_O_2_) production from neutrophils, which has previously been shown to mediate the pro-metastatic ability of neutrophils [31]. In addition, ICAM-1 promotes TNBC cells to secrete suPAR, which acts as a chemoattractant for neutrophils and facilitates tumor cell–neutrophil binding. Together, our findings suggest that ICAM-1-suPAR-CD11b axis promotes TNBC metastasis via enhancing tumor cell and neutrophil binding, which provides a strong rationale for targeting this axis to block TNBC metastasis.

## 2. Materials and Methods

### 2.1. Human Blood Analyses

All human blood analyses complied with NIH guidelines for human subject studies and were approved by the Institutional Review Boards at the Northwestern University (IRB: STU00203283) or the University of Kentucky (UK IRB #44224). The written informed consent was obtained from all subjects before blood collection. To detect CTCs, blood samples were collected in CellSave preservative tubes for CellSearch platform analyses. CellSearch Circulating Tumor Cell Kit (CellSearch, Huntington Valley, PA, USA) and PE-anti-ICAM-1 antibody (BD# 555511) were used for CTC detection following the manufacturer’s instruction. The CTCs were defined as CD45^−^, EpCAM^+^, cytokeratins^+^ (CK^+^), and DAPI^+^.

### 2.2. Animal Studies and TNBC Models

Eight- to ten-week-old female NSG (NOD SCID gamma), C57BL/6J, and CD11b^−/−^ (B6.129S4-Itgam^tm1Myd/J^) mice were purchased from Jackson Lab (Bar Harbor, ME). All mice were housed in specific pathogen-free facilities with normal chow diets and a 12:12 h light–dark cycle at 22 °C in the Division of Laboratory Animal Resources at the University of Kentucky. All animal procedures conformed to the National Institutes of Health Guide for the Care and Use of Laboratory Animals and were approved by the University of Kentucky Institutional Animal Care and Use Committee (approval number: 2019-3367).

Human PDX models were established as previously described [32]. Briefly, clinical breast cancer specimens were collected freshly, minced into bulk tumor pieces, and transplanted by trocars along with Matrigel (Corning, Corning, NY, USA) into mouse mammary fat pad areas of NSG mice. NSG mice were used for implantation of the human TNBC cell line MDA-MB-231. CD11b^−/−^ mice and C57BL/6J mice were used for implantation of mouse TNBC E0771 cells. Both MDA-MB-231 and E7001 TNBC cells were labeled with L2T, as described previously [30]. Tumor size was measured weekly using calipers. The IVIS Spectrum imaging system was used for imaging of primary tumor signals and lung metastasis in all in vivo and ex vivo experiments.

### 2.3. Cell Lines and Transfections

MDA-MB-231 and HEK-293 cell lines were obtained from ATCC (Manassas, VA, USA). E0771 cells were generously provided by Dr. Xiang Zhang, Baylor College of Medicine, Houston, TX. MDA-MB-231 and HEK-293 cell lines were cultured in DMEM high glucose, supplemented with 10% FBS and 1% penicillin–streptomycin. E0771 cell line was cultured in RPMI, supplemented with 10% FBS, 1% penicillin–streptomycin, and 10 mM HEPES. For transient knockdown, breast cancer cells were transfected with ON-TARGETplus siRNAs using Dharmafect (Dharmacon, Lafayette, CO, USA). Cells were collected for experiments 2 days after the transfection. For stable knockdown, lentiviral transduction particles containing verified MISSION shRNA constructs targeting ICAM-1 or uPAR and MISSION non-target shRNA control pLKO.1-puro plasmid vectors (Sigma-Aldrich, St. Louis, MO, USA) were packaged in HEK-293 cells. MDA-MB-231 or E0771 cells were infected by the particles, puromycin was added, and resistant colonies were selected and grown. Knockdown efficiency was confirmed by quantitative reverse transcription polymerase chain reaction (RT-PCR) and Western blotting.

### 2.4. Mouse Neutrophil Isolation

Neutrophils were purified from the bone marrow (BM) of C57BL/6J mice. Briefly, muscles and residue tissues surrounding the mouse femur and tibia were removed. The bone marrow cells were flushed with ice-cold PBS, and then centrifuged at 500× *g* for 5 min at 4 °C. After the red blood cells were removed by RBC lysis buffer, neutrophils were purified by negative selection using mouse Neutrophil Isolation Kit (Miltenyi Biotec, Gaithersburg, MD, USA) following the manufacturer’s instructions. The purity of isolated neutrophils was analyzed by flow cytometry, and cells with ≥95% purity were used for the subsequent procedures.

### 2.5. In Vitro Tumor Cell and Neutrophil Binding Assay

Tumor cells were seeded in a 24-well or 96-well plate overnight. BM neutrophil was isolated from wild-type (WT) and CD11b^−/−^ C57BL/6 mice, then neutrophils were labeled with CFSE (Thermo Fisher, Waltham, MA, USA) and co-cultured with tumor cells for 1–2 h. Suspended neutrophils were removed by washing with PBS at least three times. The remaining neutrophils associated with tumor cells were imaged by fluorescence microscopy and counted as the number of cells/fields of view.

### 2.6. Neutrophil Chemotaxis Assay

Neutrophil chemotaxis assay was analyzed using the ChemoTx^®^ Disposable Chemotaxis System with a 5 μm porous membrane (Neuro Probe, Gaithersburg, MD, USA). RPMI medium with or without murine recombination uPAR protein (10 ng/mL, SinoBiological, Chesterbrook, PA, USA) was added to the lower chambers. CFSE-labeled neutrophils were then added to the upper chamber and incubated for 2 h (37 °C, 5% CO_2_) to allow migration. The upper chambers were then carefully washed twice with fresh RPMI to remove any remaining neutrophils. To dislodge any migrated cells adherent to the underside of the filter membrane, the plate with the filter attached was centrifuged (350× *g* for 10 min). The filter was removed, and neutrophils in the lower chambers were imaged by fluorescence microscopy and counted as the number of cells/fields of view.

### 2.7. Extracellular H_2_O_2_ Production Measurement and Nicotinamide Adenine Dinucleotide Phosphate (NADPH) Oxidase Activity Assay

BM neutrophils were isolated from CD11b^−/−^ and WT C57BL/6 mice. For extracellular H_2_O_2_ production measurement, the neutrophils were suspended in RPMI medium, and H_2_O_2_ production from neutrophils was measured by Amplex™ Red Hydrogen Peroxide/Peroxidase Assay Kit (Thermo Fisher, Waltham, MA, USA) following the manufacturer’s instructions (Life Technologies, Carlsbad, CA, USA). Absorbance (OD_560_ nm) was measured in kinetic mode for 60 min using the Promega™ GloMax^®^ Plate Reader (Madison, WI, USA). NADPH oxidase (NOX) activity was measured by NADPH Oxidase Activity Assay Kit (Abcam, Waltham, MA, USA) following the manufacturer’s instructions. Briefly, neutrophils (1.5 × 10^6^) were homogenized on ice using a tissue homogenizer, and then the supernatant was collected for the assay. The absorbance (OD_600_ nm) was measured in kinetic mode for 60 min at 25 °C. NADPH oxidase activity (mU/mg) was calculated using the formula B/ΔT × P, where B is the product amount from the NADPH Oxidase Standard Curve (nmol), ΔT is the difference between Time 2 and Time 1 (min), and P is the amount of protein in the sample (mg).

### 2.8. Generation of Conditional Medium and Cytokine Array

Tumor cells were cultured overnight, and then fresh medium was added. The conditioned medium was collected after 48 h, filtered by a 0.45 µm syringe filter, and kept at –80 °C for future use. Conditioned medium from MDA-MB-231 control and ICAM-1 knockdown cells was analyzed using the Human Cytokine Array C7 (RayBiotech, Norcross, GA, USA), according to manufacturer’s instructions. Array measurements of individual factors by ELISA confirmed the validity of the assay, as shown for uPAR.

### 2.9. Enzyme-Linked Immunosorbent Assay (ELISA)

Both human and mouse blood (via cardiac puncture) were collected into EDTA-coated tubes. Plasma was separated by centrifuge at 500× *g* for 15 min at 4 °C. The levels of uPAR in conditioned medium, mouse plasma, or patient’s plasma were determined by ELISA kits (MyBioSource, San Diego, CA, USA; Boster, Pleasanton, CA, USA), according to the manufacturer’s instructions.

### 2.10. Real-Time Quantitative PCR (qRT-PCR)

Total RNA was extracted using RNeasy Mini Kit (Qiagen, Germantown, MA, USA). RNA was reverse transcribed to cDNA by High Capacity cDNA Reverse Transcription Kit (Thermo Fisher, Waltham, MA, USA). qRT-PCR was performed on a QuantStudio Real-Time PCR system (Thermo Fisher, Waltham, MA, USA) with FastStart Universal SYBR Green Master (Rox) (Sigma-Aldrich, St. Louis, MO, USA). Relative mRNA expression was calculated using the 2^–∆∆Ct^ method and normalized to GAPDH levels. Primer sequences utilized in this study are: human ICAM-1: forward 5′ ATGCCCAGACATCTGTGTCC 3′ and reverse 5′ GGGGTCTCTATGCCCAACAA 3′; human uPAR: forward 5′ TGTAAGACCAACGGGGATTGC 3′ and reverse 5′ AGCCAGTCCGATAGCTCAGG 3′; human GAPDH: forward 5′ ACAACTTTGGTATCGTGGAAGG 3′ and reverse 5′ GCCATCACGCCACAGTTTC 3′; mouse ICAM-1: forward 5′ GTGATGCTCAGGTATCCATCCA 3′ and reverse 5′ CACAGTTCTCAAAGCACAGCG 3′; mouse uPAR: forward 5′ CAGAGCTTTCCACCGAATGG 3′ and reverse 5′ GTCCCCGGCAGTTGATGAG 3′; mouse GAPDH: forward 5′ AGGTCGGTGTGAACGGATTTG 3′ and reverse 5′ TGTAGACCATGTAGTTGAGGTCA 3′. All primers and reaction conditions are summarized in Appendix A.

### 2.11. Western Blotting

Cells were washed twice with cold PBS and then lysed in RIPA buffer with a protein inhibitor cocktail (Thermo Fisher, Waltham, MA, USA) on ice for 30 min. Equal amounts of protein of each sample were run on an SDS-PAGE gel and transferred to nitrocellulose membranes. The membranes were then blocked with 2% BSA/PBS for 1 h at room temperature (RT), and then incubated with primary antibodies for 4 °C overnight and horseradish peroxidase (HRP)-conjugated secondary antibodies for 1 h at RT. The primary antibodies that were used include ICAM-1 (Proteintech, San Diego, CA), uPAR (Proteintech, San Diego, CA, USA), and actin (Thermo Fisher, Waltham, MA, USA). The primary antibodies and their dilution ratios are summarized in Appendix A.

### 2.12. Immunohistochemistry Staining

Tumor or lung paraffin-embedded sections were deparaffinized, rehydrated, pretreated by boiling in citrate buffer (pH 6.0, Vector Lab, Newark, CA, USA) for antigen retrieval, and endogenous peroxidase activity was blocked with 3% H_2_O_2_ (Sigma-Aldrich, St. Louis, MO, USA) for 10 min at RT. After blocking for 1 h at RT in blocking buffer (2% BSA in PBS), Avidin/Biotin Blocking Kit (Vector Lab, Newark, CA, USA) was applied to block all endogenous biotin, biotin receptors, and avidin binding sites present in tissues. The sections were then incubated with anti-Ly6G, a specific marker for mouse neutrophils (BD Biosciences, Franklin Lakes, NJ, USA), in blocking buffer overnight at 4 °C, followed by biotinylated secondary antibody for 1 h. Finally, peroxidase substrate diaminobenzidine (Vector Lab, Newark, CA, USA) was applied. The slides were rinsed and counterstained with hematoxylin. Mounting solution and coverslips were added. Images were acquired with Aperio ScanScope XT high-throughput digital slide scanner system.

### 2.13. Bioinformatics Analysis

The correlation of PLAUR gene expression with neutrophil infiltration level in The Cancer Genome Atlas (TCGA) was analyzed. The breast cancer-basal subtype (*n* = 191; *p* < 0.05) was analyzed using the tumor immune estimation resource with tumor purity adjustment (Timer2.0, http://timer.cistrome.org; accessed on 15 July 2022) [33]. The protein expressions in different breast cancer subtypes (*p* < 0.05) were retrieved from the Clinical Proteomic Tumor Analysis Consortium (CPTAC) datasets using UALCAN (The University of Alabama at Birmingham Cancer Data Analysis Portal, http://ualcan.path.uab.edu/index.html; accessed on 20 July 2022) [34]. The PLAUR and ICAM-1 expression in the TCGA-BRCA dataset and corresponding normal control tissues were evaluated via GEPIA 2 (Gene Expression Profiling Interactive Analysis, http://gepia2.cancer-pku.cn; accessed on 18 July 2022); (log2FC) > 1, adjusted *p*-value < 0.01) [35]. In addition, PLAUR and ICAM-1 correlation in TCGA-BRCA (*p* < 0.05) was analyzed using Pearson’s correlation coefficient via GEPIA 2. The overall survival and distant metastasis-free survival of breast cancer patient cohorts in publicly or selectively available databases were analyzed via Prognoscan (http://www.prognoscan.org; accessed on 1 July 2022) [36].

### 2.14. Statistical Analysis

Student’s *t*-test was performed for the statistical analyses between two samples as appropriate using GraphPad Prism software ver. 9.2 One-way ANOVA (followed by Tukey post hoc test) was performed to analyze differences among multiple groups. Data are presented as mean ± SD from at least three replicates, and *p* < 0.05 was considered significant.

## 3. Results

### 3.1. ICAM-1 and CD11b Mediate Tumor Cell and Neutrophil Binding

Neutrophils are the most abundant white blood cells in our body. Interestingly, we found that neutrophils (identified by the unique fragmented shape of the neutrophil nucleus) are frequently bound with CTCs forming neutrophil–CTC clusters in several breast cancer PDX models (Figure 1A). Using the FDA-approved CellSearch platform (Figure 1B), we also detected CTC (defined as EpCAM^+^CK^+^DAPI^+^CD45^−^)–neutrophil clusters in metastatic breast cancer patients (Figure 1B). Recently, we found that ICAM-1 expression was increased by 200-fold in lung metastases, compared to primary tumor cells in several TNBC PDX models [30]. Given that vascular cell adhesion molecule 1 (VCAM-1) was identified as a molecule mediating CTC–neutrophil cluster formation [14], we explored whether adhesion molecule ICAM-1 could also be involved in CTC-neutrophil formation. Indeed, we found that some ICAM-1^+^ CTCs were associated with neutrophils in breast cancer patients (Figure 1C), suggesting that ICAM-1 on tumor cells may directly mediate tumor cells binding with neutrophils. To test this hypothesis, we performed in vitro tumor cell and neutrophil binding assay. The CFSE-labeled neutrophils isolated from wild-type (WT) C57BL/6 mice were co-cultured with either E0771 or E0771 ICAM-1 knockdown (shICAM-1) cells. After removal of the unbound neutrophils, the remaining neutrophils were imaged and counted. We found that ICAM-1 knockdown in E0771 cells significantly reduced their binding with WT neutrophils (control: 140 ± 10; shICAM-1: 85 ± 5; *p* < 0.05) (Figure 1D,E). It is known that CD11b on activated neutrophils can bind to ICAM-1 on endothelial cells to mediate firm adhesion to endothelial cells during acute and chronic inflammatory diseases [37,38,39,40]. We wondered whether ICAM-1^+^ tumor cell and neutrophil binding is also dependent on CD11b. Indeed, the CD11b^−/−^ neutrophils isolated from CD11b^−/−^ mice displayed much lower binding ability (40 ± 10; *p* < 0.05) with E0771 cells compared with WT neutrophils (Figure 1D,E). To further confirm the role of ICAM-1 and CD11b in mediating tumor cells and neutrophils binding, we sorted ICAM-1^+^ and ICAM-1^−^ E0771 cells and tested their binding ability with CD11b^−/−^ neutrophils. We found that ICAM-1^+^ tumor cells showed much higher binding ability with CD11b^−/−^ neutrophils than ICAM-1^−^ tumor cells (Appendix A). Taken together, these data suggest that ICAM-1 on tumor cells binds to CD11b on neutrophils, thereby promoting a firm binding of neutrophils with tumor cells.

### 3.2. CD11b Deficiency Inhibits TNBC Metastasis

CTC-associated neutrophils enhance survival and metastatic ability of CTCs [14,41]. Since CD11b mediates the binding of neutrophils with tumor cells (Figure 1D,E and Appendix A), it raises a possibility that CD11b may play a role in metastasis. Given the previous studies have demonstrated that neutrophils primarily impact CTCs in the post-intravasation processes of metastasis [42], we injected the L2T-labeled (expressing both luciferase and tdTomato red fluorescent protein) E0771 cells (1 × 10^6^) into WT and CD11b^−/−^ C56BL/6 mice via tail vein to test the effect of CD11b deficiency on the post-intravasation step of metastasis. Using bioluminescence imaging (BLI), we found that cells colonized in the lungs were dramatically reduced in CD11b^−/−^ mice compared with WT mice 24 h after injection of tumor cells (Figure 2A,B). Consequently, the lung metastases were significantly decreased in CD11b^−/−^ mice at 25 days (D25) after intravenous injection of tumor cells (*n* = 5, *p* < 0.05) (Figure 2C). Furthermore, when the E0771 cells (1 × 10^6^) were injected into the mammary fat pads of the mice, spontaneous lung metastasis was significantly reduced in CD11b^−/−^ mice (*n* = 5, *p* < 0.05), but there was no significant difference in primary tumor growth between WT or CD11b^−/−^ C57BL/6 mice (Figure 2D,E). Taken together, these data suggest that CD11b deficiency most likely inhibits metastasis after tumor cell intravasation.

NK cells are the main immune cells in the blood to kill CTCs [42,43]. CTC clusters exhibit higher resistance to NK cells [44]. In addition, CTC-associated neutrophils enhance survival of CTCs [14,41]. However, the mechanism is unclear. Recent studies have demonstrated that neutrophils suppress the tumoricidal activity of NK cells by producing reactive oxygen species (ROS) to promote breast cancer lung metastasis [31]. Given that CD11b is involved in ROS production in macrophages [45], we tested whether CD11b could regulate ROS production from neutrophils. The BM neutrophils were isolated from WT or CD11b^−/−^ C57BL/6 mice, then the extracellular H_2_O_2_ production was measured using Amplex™ Red Hydrogen Peroxide/Peroxidase Assay Kit over time. Compared with WT neutrophils, the H_2_O_2_ production from CD11b^−/−^ neutrophils was significantly reduced (Appendix A). The majority of ROS/H_2_O_2_ in neutrophils is generated by the activation of the NADPH oxidases (NOX) [46,47]. To further determine how CD11b regulates H_2_O_2_ production of neutrophils, we compared NOX activity between WT and CD11b^−/−^ neutrophils and found that NOX activity was lower in CD11b^−/−^ neutrophils than WT neutrophils (Appendix A). Collectively, these data suggest that CD11b mediates NOXs activation to produce H_2_O_2_ in neutrophils, which could inhibit NK cell’s tumoricidal activity to facilitate CTC survival.

### 3.3. ICAM-1 Promotes TNBC Cells to Release suPAR, which Functions as a Chemoattractant for Neutrophils

Neutrophils can sense extracellular chemical gradients and exhibit directional migration toward higher concentrations in a process termed chemotaxis [48,49]. This directed neutrophil recruitment is orchestrated by chemoattractants such as chemokines (cytokines with chemotactic activities) and lipids [50]. Therefore, we expect that ICAM-1^+^ tumor cells may secrete specific chemoattractants to recruit neutrophils, which then facilitate their binding with neutrophils. To test this hypothesis, we performed cytokine array analysis using conditioned medium from cultured control and ICAM-1 knockdown MDA-MB-231 cells. Among 61 cytokines, uPAR in the conditioned medium collected from ICAM-1 knockdown MDA-MB-231 cells was dramatically reduced compared with control cells (*p* < 0.05) (Figure 3A), which was further confirmed by uPAR ELISA assay (*p* < 0.05) (Figure 3B). ICAM-1 knockdown also decreased uPAR protein levels (Figure 3C,D and Appendix A), but not the uPAR mRNA levels (Appendix A), suggesting that ICAM-1 regulates both uPAR release and expression.

The suPAR levels are increased in the blood of cancer patients and are associated with poor patient prognosis in several types of cancers [22,23,24,25,26]. However, how suPAR levels in the blood are increased is unclear. We found that plasma levels of uPAR were significantly elevated in both MDA-MB-231 tumor-bearing NSG mice and E0771 tumor-bearing C57BL/6 mice (Figure 3E,F). However, the uPAR plasma levels were dramatically reduced in mice bearing uPAR knockdown tumor cells (Figure 3E,F). Consistently, the uPAR plasma levels were higher in patients whose primary tumors had not been resected than those whose primary tumor was resected (Figure 3G). These data suggest that tumor cell-secreted suPAR contributes to the elevated uPAR levels in the blood. Of note, both plasma levels of human uPAR (secreted by human MDA-MB-231 TNBC cells) and mouse uPAR (secreted by the host) were elevated in the MDA-MB-231-bearing NSG mice as measured by human uPAR and mouse uPAR ELISA kits, respectively (Figure 3E), indicating that in addition to tumor cells, the host cells also secrete uPAR, contributing to the elevated plasma levels of uPAR.

Next, we examined whether suPAR could act as a chemoattractant for neutrophils. To address this, we performed a chemotaxis assay, and found that addition of suPAR (10 ng/mL) in the lower chamber significantly increased the migration of CFSE-labeled mouse neutrophils from the upper chamber to the lower chamber, compared with the control group (*p* < 0.01,) (Figure 4A,B). Furthermore, when control or uPAR knockdown cells (generated by lentiviral vector-mediated transduction of uPAR short hairpin RNA) were injected into the mice, the lung-infiltrated neutrophils (identified by IHC staining using anti-Ly6G antibody) were also dramatically reduced in mice-bearing uPAR knockdown cells compared with the control group in both E0771 model and MDA-MB-231 model (Figure 4C,D), further suggesting that uPAR promotes neutrophil recruitment. In line with these findings, the uPAR expression is also positively correlated with neutrophil infiltration level in the TCGA breast cancer-basal subtype (*p* < 0.05; *n* = 191) (Figure 4E). Taken together, these findings reveal a novel role for suPAR as a chemoattractant for neutrophils.

### 3.4. uPAR Knockdown Inhibits TNBC Metastasis

Since suPAR functions as a chemoattractant for neutrophils, which can facilitate tumor cell and neutrophil binding, it raises a possibility that uPAR may promote TNBC metastasis. To test this possibility, L2T-labelled uPAR knockdown (shuPAR) MDA-MB-231 and E0771 cells were generated. The efficiency of uPAR knockdown was validated by the reduced uPAR mRNA and protein levels in both cell lines (Appendix A). In addition, the suPAR level in the supernatant of cultured uPAR knockdown cells was also decreased (*p* < 0.05) (Appendix A). To determine the role of uPAR in metastasis, we subcutaneously injected L2T-labelled control or uPAR knockdown MDA-MB-231 and E0771 cells close to the fourth mammary fat pad areas of NSG and C57/BL6 mice, respectively. Although there were no significant differences in tumor growth between control and shuPAR groups (Figure 5B–D), spontaneous lung metastases (measured between 42 and 45 days after tumor implantation) were dramatically reduced in uPAR knockdown groups (*p* < 0.05; *n* = 5/group) (Figure 5A–D), suggesting that uPAR mainly plays a role in the post-intravasation processes of metastasis but not the primary tumor growth. To further confirm the role of uPAR in promoting metastasis after post-intravasation of tumor cells, we directly injected control and uPAR knockdown MDA-MB-231 and E0771 cells into the blood of mice via tail vein. Consistently, the lung metastasis was also inhibited by uPAR knockdown (Figure 5E,F). Collectively, these data strongly suggest that uPAR plays an important role in promoting TNBC metastasis.

### 3.5. High uPAR Expression Correlates with Worse Breast Cancer Patient Outcome

To evaluate the potential clinical association of ICAM-1 and uPAR in breast cancer patients, we first compared ICAM-1 and uPAR protein expression between different breast cancer subtypes using Clinical Proteomic Tumor Analysis Consortium (CPTAC) data sets. We found that the protein levels of both ICAM-1 and uPAR were significantly higher in TNBC patients compared with other subtypes (Figure 6A,B). In addition, the uPAR plasma levels were also higher in TNBC patients compared with other subtypes of breast cancers (Figure 6C). Consistent with mouse studies, ICAM-1 gene expression was also positively correlated with PLAUR gene expression in TCGA breast cancer data sets (*p* < 0.05) (Figure 6D). Furthermore, PLAUR was highly expressed in breast cancer tissues compared with matched TCGA normal mammary tissues (Log2FC > 1, *p* < 0.01) (Figure 6E). Lastly, we evaluated the clinical relevance of uPAR with breast cancer patient survival. Using PrognoScan analyses [35], we observed that high level of PLAUR mRNA expression in breast tumors was associated with poor distant metastasis-free survival (DMFS) and overall survival (OS) (Figure 6F,G). Taken together, these data indicate that high uPAR expression is clinically associated with worse breast cancer patient outcome, particularly TNBC patients.

## 4. Discussion

We previously reported that homotypic CTC clusters were present in several PDX breast cancer mouse models and breast cancer patients [10]. We also identified ICAM-1 as a new driver for TNBC metastasis by mediating homotypic CTC cluster formation [30]. In this study, we found that some CTCs also formed heterotypic clusters with neutrophils. Since neutrophils enhance the metastatic ability of CTCs by interacting with them [14], we further investigated the mechanism underlying CTC–neutrophil cluster formation in TNBC models, with the hope of discovering novel therapeutic targets and strategies to treat metastatic TNBC, which has limited treatment options.

Neutrophils are the most abundant immune cells in the blood circulation. Accumulating evidence indicates that neutrophils play an important role in metastasis [51,52]. Interestingly, neutrophils play either pro-metastatic or anti-metastatic effects which are governed by the NK cell status of the host [31]. In the presence of NK cells, neutrophils promote metastasis. However, when NK cells are absent, neutrophils inhibit metastasis. It is reported that most CTCs are killed by NK cells in the circulation [42,43]. Since neutrophils can suppress the tumoricidal activity of NK cells by producing ROS [31], it is most likely that CTC-associated neutrophils could also prevent CTCs from being attacked by NK cells via ROS production. Indeed, we found that CD11b on neutrophils not only mediates tumor cell and neutrophil binding, but also is required for H_2_O_2_ production from neutrophils. Furthermore, lung metastasis generated by TNBCs was significantly reduced in the CD11b-deficient host mice, although the primary tumor growth was not significantly changed, suggesting CD11b plays an important role in promoting metastasis after tumor cell intravasation. In addition to neutrophils, CD11b expresses on the surface of other myeloid cells, such as monocytes and macrophages. Studies have shown that inhibition of CD11b accelerates tumor growth by promoting immune suppressive macrophage polarization, although the effect of CD11b inhibition on metastasis was not investigated in the study [53]. We did not detect a significant difference in primary tumor growth between WT and CD11b-deficient mice, but we noticed that the tumors grow faster at the early time point in CD11b-deficient mice. Therefore, it is possible that CD11b may play an opposite role at the different stages of tumor progression and metastasis. It is also likely that the effect of CD11b is cell-type dependent. Since neutrophil-specific CD11b knockout mice are not currently available, generating CD11b conditional knockout mice or bone marrow chimeric mice between CD11b knockout and wild-type mice could be critical to address these questions in future studies.

Under homeostatic conditions, neutrophils have a short lifespan (less than 24 h) in circulation and undergo spontaneous apoptosis [54,55]. Recent studies demonstrated that neutrophil lifespan is significantly extended to several days in cancer setting [56,57], although the mechanism is not fully understood. Interestingly, we found that CD11b on neutrophils was upregulated in TNBC-bearing mice, which is directly mediated by secretome from tumor cells. Furthermore, CD11b upregulation mediated prolonged survival of neutrophils (unpublished data). Our ongoing studies focus on understanding how tumor cells upregulate CD11b expression on neutrophils to prolong their survival, and the role of prolonged neutrophil survival in TNBC metastasis. Since CD11b mediates trans-endothelial migration of neutrophils [29], it is possible that CTCs upregulate CD11b expression on neutrophils, which in turn enhances the adhesion of neutrophils with endothelial cells to promote trans-endothelial migration of neutrophil- CTC cluster into metastatic organs. We will test this possibility in our future studies.

PAR is a glycoprotein composed of three homologous domains (D1, D2, and D3) [58] and can be shed from the cell surface as an entire uPAR protein by phospholipase D [59], or cleaved by proteases to release cleaved suPAR [60]. It is known that PLD activity is dependent on the activation of protein kinase C (PKC) [61,62]. It is also known that ICAM-1 can activate PKC [63]. It is likely that the PKC–PLD pathway may be involved in ICAM-1-mediated secretion of uPAR, which warrants further investigation. uPAR is expressed at high levels in many human cancers and correlates with a poor prognosis and early invasion and metastasis. In addition, suPAR is detected in the blood of cancer patients, and increased suPAR level is associated with poor patient prognosis in several types of cancers [22,23,24,25,26]. However, how suPAR is elevated in cancer patients is poorly understood. uPAR is expressed at low levels in healthy tissues, mainly on the cell membrane of immune cells, such as monocytes, neutrophils, and activated T-lymphocytes [17]. However, inflammatory stimulation induces the upregulation of uPAR expression on immune cells, consequently elevating suPAR levels, which have been observed in several inflammatory diseases [64]. Interestingly, we found that plasma levels of mouse suPAR were increased in MDA-MB-231-bearing NSG mice, in addition to human suPAR. Since MDA-MB-231 is a human breast cancer cell line, the increased mouse suPAR is derived from the mouse host, suggesting that tumors may induce an inflammatory response in these mouse models, which upregulates uPAR on the immune cells, and elevates endogenous mouse suPAR levels. In a limited number of breast cancer patients, we found that if patient’s primary tumors have not been resected, they have higher plasma suPAR levels compared with patients whose primary tumors have been resected, indicating tumor-secreted suPAR may be a major source for elevated plasma levels of suPAR in these patients. Since it is impossible to distinguish the tumor cell or immune cell-derived suPAR in human patients, measurement of the uPAR expression in the primary tumors combined with suPAR levels in the blood may be necessary to better understand the role of uPAR in cancers. Since inflammation is a hallmark of cancer, it is important to understand whether tumor cell-derived and host cell-derived suPAR play the same or different roles in tumor progression and metastasis in future studies. It is also interesting to look at whether suPAR levels correlate with the general inflammation markers such as C-reactive protein (CRP), neutrophil count, and the neutrophil-lymphocyte ratio (NLR) in cancer patients, which may help to predict whether elevated suPAR is mainly derived from tumor cells or immune cells.

Immune checkpoint inhibitors (ICIs) have revolutionized treatment in a broad spectrum of metastatic cancers, and the combination of an anti-PD1 inhibitor with chemotherapy has recently emerged as a novel treatment option for both early and metastatic TNBC [65,66]. However, only a small subset of patients shows clinical benefit, and patients often develop resistance, highlighting a strong need to discover novel biomarkers to better predict patient response and develop new strategies to overcome resistance. Accumulation of immunosuppressive myeloid cells, including neutrophils, is one of the top reasons for resistance to ICIs. Since we discovered that suPAR functions as a new neutrophil chemoattractant, and is correlated with neutrophil infiltration, whether uPAR expression levels on tumor cells and suPAR in blood could be the new biomarkers to facilitate the prediction of patient response to ICIs warrants further investigation. A recent preclinical study found that a combination of anti-uPAR and anti-PD1 remarkably inhibits tumor growth and prolongs survival in diffuse-type gastric cancer [67]. Further study is needed to evaluate whether blocking uPAR could be a novel strategy to improve anti-PD1 efficacy in TNBC.

## 5. Conclusions

In summary, we demonstrated that ICAM-1 and CD11b mediate tumor cell and neutrophil binding. In the meantime, CD11b on neutrophils promotes H_2_O_2_ production from neutrophils, which suppresses NK cell-mediated tumor cell killing [31]. In addition, ICAM-1 promotes TNBC cells to secrete suPAR, which functions as a neutrophil chemoattractant to facilitate tumor cell and neutrophil binding (Figure 6H). Collectively, our findings reveal a new mechanism of CTC–neutrophil cluster formation via ICAM-1-suPAR-CD11b axis-mediated tumor cell and neutrophil binding, which could represent new therapeutic targets for treating metastatic TNBC.

## Figures and Tables

**Figure 1 cancers-15-02734-f001:**
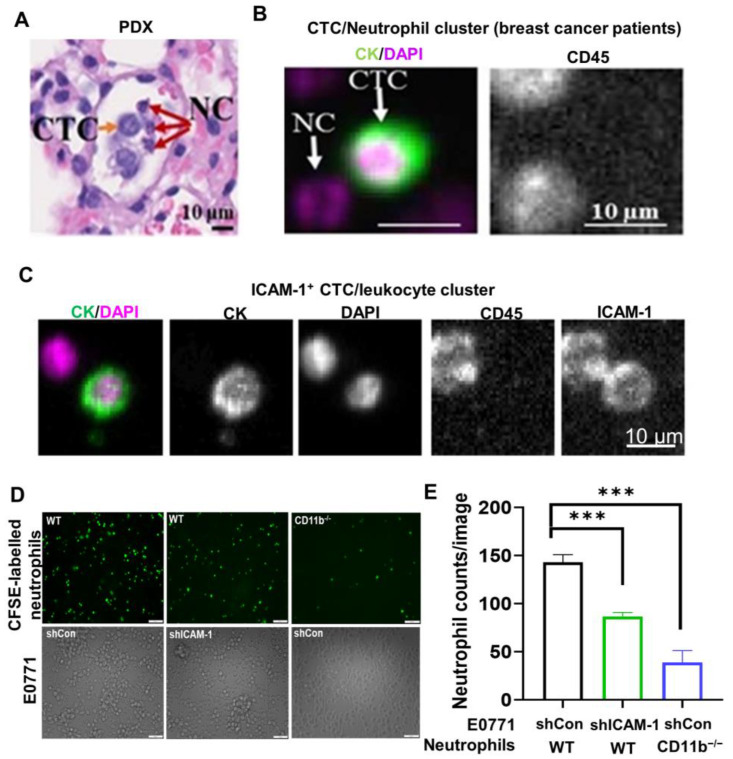
ICAM-1 and CD11b mediate tumor cell and neutrophil interaction. (**A**) Representative H&E staining of lung sections shows a CTC–neutrophil cluster (neutrophils are identified by the morphology with segmented nuclei, red arrow pointed) in the lung vasculature of the TNBC PDX model. (**B**) Representative images of a neutrophil (segmented nuclei and CD45^+^)–CTC (EpCAM^+^CK^+^DAPI^+^CD45^−^) cluster detected in a breast cancer patient by the CellSearch platform. (**C**) Representative images of an ICAM-1^+^ CTC–leukocyte (DAPI^+^CD45^+^) cluster detected in a breast cancer patient by the CellSearch platform. (**D**,**E**) In vitro binding assay shows ICAM-1 on tumor cells and CD11b on neutrophils are important for tumor cell and neutrophil binding. BM neutrophils isolated from WT and CD11b^−/−^ mice were labeled with CFSE, and co-cultured with control (shCon) or ICAM-1 knockdown (shICAM-1) E0771 cells for 1 h. The binding of neutrophils with tumor cells was imaged (**D**) and quantitated (**E**) (One-way ANOVA, *** *p* < 0.001, *n* = 5).

**Figure 2 cancers-15-02734-f002:**
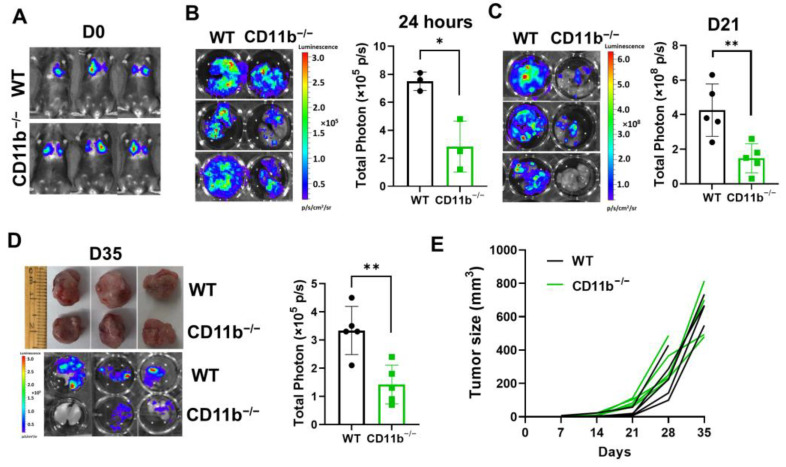
CD11b deficiency inhibits TNBC metastasis. (**A**) Bioluminescence imaging (BLI) of WT and CD11b^−/−^ C57BL/6 mice at 2 h after injection of L2T-labelled E0771 cells (1 × 10^6^) via tail vein. (**B**) Ex vivo lung BLI images (left) and quantitative data (right) of WT and CD11b^−/−^ C57BL/6 mice at 24 h after injection of tumor cells via tail vein (*t*-test, * *p* < 0.05, *n* = 3). (**C**) Representative ex vivo lung BLI images (left) and quantitative data (right) of WT or CD11b^−/−^ C57BL/6 mice at 21 days after injection of tumor cells via tail vein (*t*-test, ** *p* < 0.05, *n* = 5). (**D**) Representative tumor and ex vivo lung BLI images (left) and quantitative data (right) of lung metastasis of WT or CD11b^−/−^ C57BL/6 mice at 35 days (D35) after subcutaneous injection of L2T-labelled E0771 cells (1 × 10^6^) to the fourth mammary fat pad areas (*t*-test, ** *p* < 0.01, *n* = 5). (**E**) Tumor growth of individual tumor of WT (black) and CD11b^−/−^ (green) C57BL/6 mice from D was measured at indicated time points after tumor implantation (*n* = 5/group).

**Figure 3 cancers-15-02734-f003:**
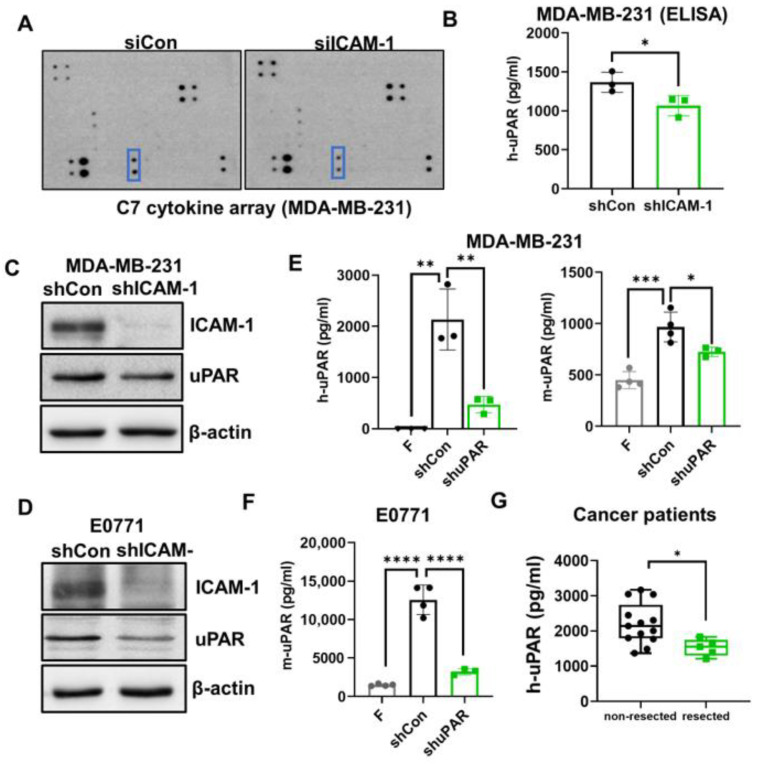
ICAM-1 promotes TNBC cells to secrete suPAR. (**A**) C7 human cytokine array was used to detect the cytokines in the supernatants of cultured control (siCon) and ICAM-1 knockdown (siICAM-1) MDA-MB-231 cells. uPAR is indicated in the blue boxes. (**B**) The secreted human uPAR (h-uPAR) in the supernatants from cultured control (shCon) and ICAM-1 knockdown (shICAM-1) MDA-MB-231 cells was measured by a human uPAR ELISA kit (*t*-test, * *p* < 0.05, *n* = 3). (**C**,**D**) ICAM-1 knockdown (shICAM-1) reduced uPAR expression in (**C**) MDA-MB-231 and (**D**) E0771 cells determined by Western blotting. (**E**) The plasma levels of human and mouse uPAR in tumor-free (F), control (shCon) and uPAR knockdown (shuPAR) MDA-MB-231 tumor-bearing mice were measured by human and mouse uPAR ELISA kits, respectively (*t*-test, * *p* < 0.05, ** *p* < 0.01, *** *p* < 0.001, *n* = 3–5). (**F**) The plasma levels of mouse uPAR in tumor-free (F), control (shCon) and uPAR knockdown (shuPAR) E0771 tumor-bearing mice were measured by a mouse uPAR ELISA kit (*t*-test, **** *p* < 0.0001, *n* = 4). (**G**) The plasma levels of uPAR in breast cancer patients with primary tumors (non-resected) or without primary tumors (resected) were measured by human uPAR ELISA kits (Welch’s *t* test, * *p* < 0.05, *n*= 13 non-resected, *n* = 5 resected).

**Figure 4 cancers-15-02734-f004:**
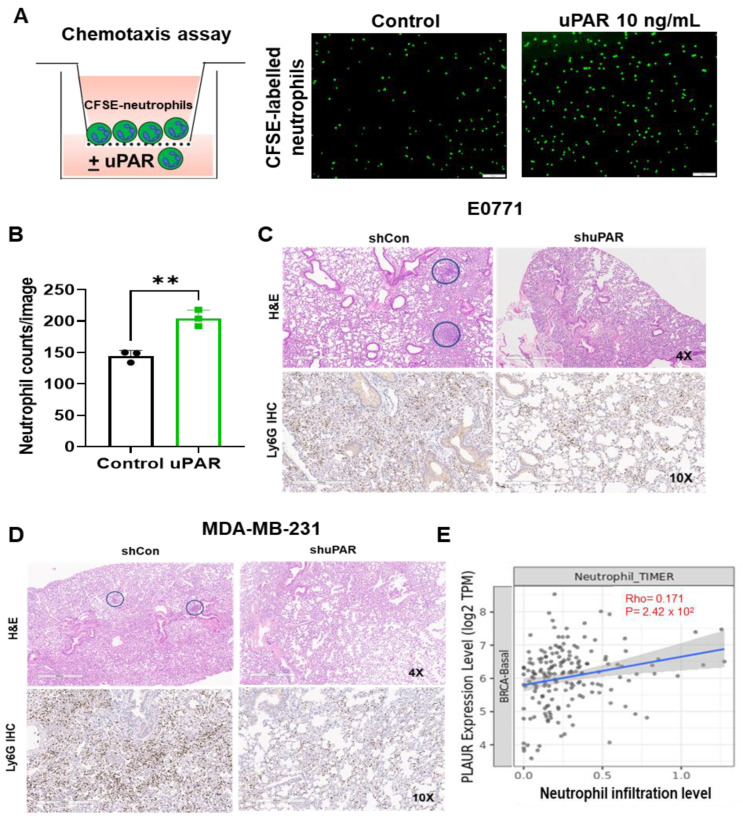
uPAR is a chemoattractant for neutrophils. (**A**,**B**) Chemotaxis assay demonstrates that uPAR promotes neutrophil migration. Neutrophils were isolated from the bone marrow of wild-type C57BL/6 mice and labeled with CFSE. The migration of neutrophils from the upper chamber to the lower chamber (RPMI with or without 10 ng/mL uPAR) was imaged (**A**) and quantitated (**B**) as described in Materials and Methods (*t*-test, ** *p* < 0.01, *n* = 3). (**C**,**D**) Representative IHC staining (lower) shows the infiltration of neutrophils into the lungs (stained by anti-Ly6G antibody) is reduced in mice implanted with uPAR knockdown E0771 (**C**) and MDA-MB-231 (**D**) cells. Representative H&E staining (upper) indicates lung metastasis (in blue circle) of tumor-bearing mice. (**E**) uPAR expression positively correlates with neutrophil infiltration level in the TCGA breast cancer-basal subtype (*n* = 191; *p* < 0.05, analyzed using Timer2.0 at http://timer.cistrome.org/, accessed on 15 July 2022). The correlation was evaluated by Spearman’s correlation coefficient.

**Figure 5 cancers-15-02734-f005:**
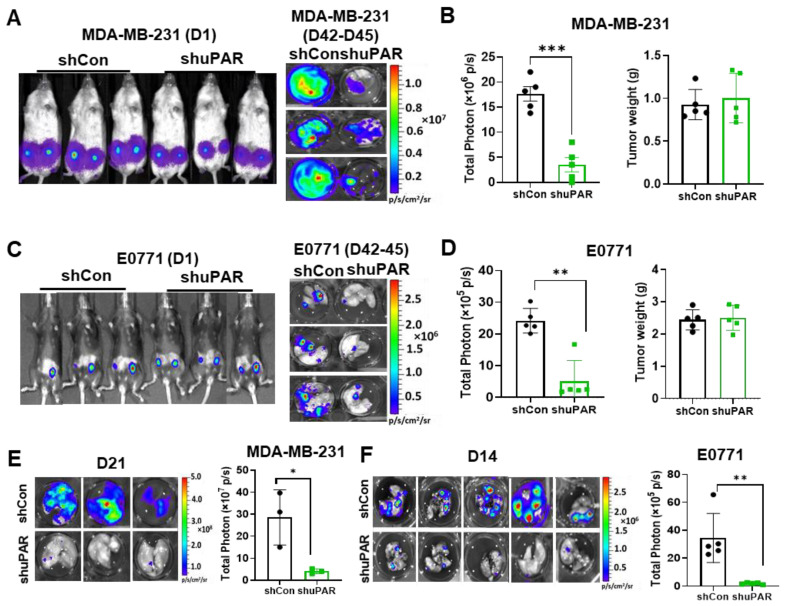
uPAR promotes lung metastasis of TNBC. (**A**) Representative BLI images of mice implanted with L2T-labelled control and uPAR knockdown MDA-MB-231 cells were taken on day 1 (D1) after tumor implantation, and ex vivo lung BLI images were taken between day 42 and day 45 after tumor implantation. (**B**) Quantification of lung metastasis (total flux photon, p/s) and tumor weight of mice from A (*t*-test, *** *p* < 0.001, *n* = 5/group). (**C**) Representative BLI images of mice implanted with L2T-labelled control and uPAR knockdown E0771 cells were taken on day 1 (D1) after tumor implantation, and ex vivo lung BLI images were taken between day 42 and day 45 after tumor implantation. (**D**) Quantification of lung metastasis (total flux photon, p/s) and tumor growth of mice from C (*t*-test, ** *p* < 0.01, *n* = 5/group). (**E**) Ex vivo lung BLI images (left) and quantification (right) of mice injected with L2T-labelled control and uPAR knockdown MDA-MB-231 cells via tail vein (*t*-test, * *p* < 0.05, *n* = 3/group). (**F**) Ex vivo lung BLI images (left) and quantification (right) of mice injected with L2T-labelled control and uPAR knockdown E0771 cells via tail vein (*t*-test, ** *p* < 0.01, *n* = 5/group).

**Figure 6 cancers-15-02734-f006:**
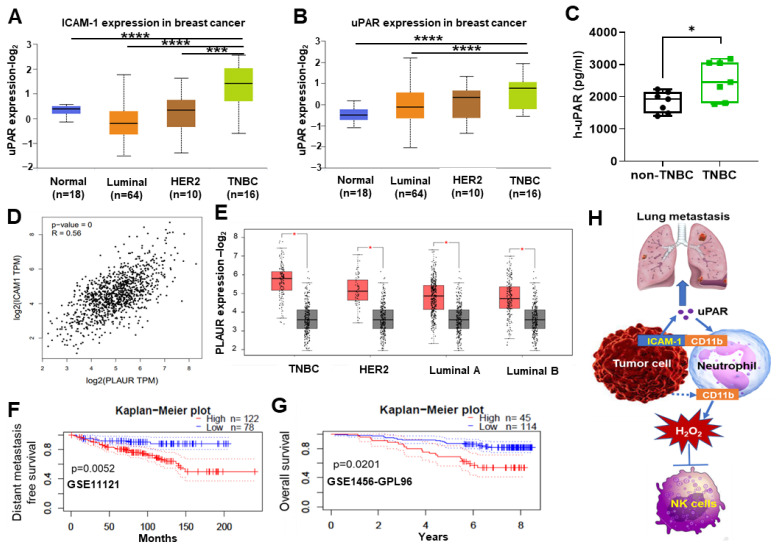
uPAR expression level is associated with clinical outcomes of breast cancer patients. (**A**,**B**) The protein expression levels of ICAM1 and uPAR in different breast cancer subtypes (*** *p* < 0.001, **** *p* < 0.0001). The graphs were retrieved from UALCAN (http://ualcan.path.uab.edu/index.html, accessed on 20 July 2022) using Clinical Proteomic Tumor Analysis Consortium (CPTAC) data sets. (**C**) The plasma levels of uPAR in non-TNBC and TNBC patients were measured by human uPAR ELISA kits (*t*-test, * *p* < 0.05, *n* = 7 non-TNBC, *n* = 7 TNBC). (**D**) The positive correlation between ICAM1 and PLAUR expression level in TCGA breast cancer data sets (Pearson’s correlation, *p* < 0.05). Data were retrieved from GEPIA2 (http://gepia2.cancer-pku.cn, accessed on 18 July 2022). (**E**) The expression level of PLAUR in different subtypes of TCGA breast cancer dataset compared with the matched TCGA normal dataset (Log2FC > 1, *p* < 0.01). Data were retrieved from GEPIA2 (http://gepia2.cancer-pku.cn, accessed on 18 July 2022). (**F**,**G**) Distant metastasis-free survival (GSE11121) and overall survival (GSE1456-GPL96) of the breast cancer patients with high and low uPAR expression in selectively available public databases were analyzed via the online program Prognoscan (http://www.prognoscan.org/, accessed on 1 July 2022). Log-rank *p* = 0.0052 and 0.0201, as indicated. (**H**) Diagram of ICAM-1-uPAR-CD11b axis in promoting TNBC metastasis via CTC-neutrophil cluster.

## Data Availability

All data supporting the findings of this study are available within the article and its Appendix A and from the corresponding author upon reasonable request.

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
