# Peer review of "ICAM-1-suPAR-CD11b Axis Is a Novel Therapeutic Target for Metastatic Triple-Negative Breast Cancer"

_cancers, 2023, doi:10.3390/cancers15102734_

Round 1

Reviewer 1 Report

* The detection of ICAM-1-suPAR-CD11b as a novel target for treating triple breast cancer is an exciting issue. The investigation is well constructed, and I have some minor problems with accepting this manuscript.

* I suggest making the title more attractive for readers by including to target triple-negative breast cancer metastasis. It is your final conclusion. right?

* Line 19: Please, define ICAM-1 at its first mention.

* Line 36: Please, define suPAR at its first mention.

* Line 103: Please, provide the approval number.

* Line 106: Please, mention the catalog number for CellSearch Circulating Tumor Cell 105 Kit and for all used chemicals and kits.

* Line 116: Please, mention the approval number.

* Line 200-207: Please, include all primers and their specific experimental conditions in a table.

* Line 214-216: Please, include all antibodies and their specific experimental conditions in a table.

* Line 217: did authors use antigen retrieval for anti-Ly6G? why did the authors use this antibody? 

* Line 223: Did the authors use of streptavidin step?

* Line 250, 257: I suggest transferring all references used in the result section to the discussion section.

* Line 299: Figure 3 is missing

Reviewer 2 Report

The authors proposed that the involvement of the ICAM-1-uPAR-CD11b axis in tumor cell-neutrophil binding and subsequent metastasis formation of triple-negative breast cancer. However, the data are not sufficient to support their assumption.

Major comments

#1. Figure 3 is lacking and therefore, it is impossible to evaluate the data at all.

#2. Animal studies. The authors should describe the implantation procedures in more detail, because even the cited paper (ref. 32) did not describe the details of the procedures.

#3. Figures 1 and 2. The authors demonstrated the data arising from using a single murine breast cancer cell line and therefore, should use additional cancer cell lines to prove the generality of their observations.

#4. Figure 2. The authors claimed the involvement of CD11b expressed in neutrophils by using CD11b knockout mice because neutrophil-specific CD11b knockout mice are not available at present. Instead of neutrophil-specific CD11b knockout mice, the authors should conduct similar experiments by using bone marrow chimeric mice between CD11b knockout and wild-type mice, in order to prove the involvement of CD11b expressed in bone marrow-derived cells.

Minor comments

#1. Figure 4B. shICAM-1 treatment decreased the amount of secreted uPAR only marginally. It is very doubtful on the biological significance of the observations. Moreover, the authors should clarify the molecular mechanisms underlying ICAM-1-mediated secretion of uPAR.

#2. Figure 5. The authors should examine whether similar observations are obtained by using human system.

#3. Figure 6A and 6C. The authors determined the tumor size at the injected site 1 day after the implantation. Determined tumor sizes reflected tumor cell viability immediately after the implantation but not tumor growth rates at the primary site. Thus, it is impossible to negate the involvement of uPAR in the primary tumor growth.

Reviewer 3 Report

In this manuscript, Li et al first found that neutrophils were able to bind to circulating tumor cells (CTC) in several breast cancer PDX models. At the same time, the expression of ICAM-1+ was found to be increased by 200-fold in lung metastasis compared to that in primary tumor cells. This led to the identification of ICAM-1+ CTC-neutrophil cluster in breast cancer. It was further found that CD11b expressed by neutrophils mediated the binding with ICAM-1 in CTC. In addition, a molecule known as uPAR expressed by tumor cells can be released from the cells to become soluble uPAR which functions as a chemoattractant to recruit neutrophils for the formation of CTC-neutrophils. Finally, it was found that higher uPAR was corelated with poorer prognosis of breast cancer patients. These findings are interesting and shed light on the pathogenesis of breast cancer metastasis. 

I would suggest the authors to address the following minor issues before publication of the manuscript:

1. Section 3.3 describes the regulation of ROS production by CD11b in neutrophils. It seems to me that this section is out of the main story. I would suggest to remove it from the manuscript or move it to supplemental materials. 

2. In Figure 2E, the tumor sizes from WT and CD11b KO should be combined into one plot for comparison.

3. Neutrophils are short-lived. Suggest the authors to discuss how long these CTC-neutrophil cluster will stay and when the neutrophils die, what will happen to the CTC cells, etc..

4. There are numerous grammar/language mistakes. Suggest the authors to find a native English speaker to help with editing the manuscript. 

Reviewer 4 Report

The study by Li et al. demonstrated that circulating tumor cells (CTCs) especially triple-negative breast cancer (TNBC) cells in this manuscript promote metastasis through ICAM-1 to secret suPAR and attract/bind neutrophils in a CD11b-dependent manner. The study is overall interesting, however, a few points need to be addressed to ensure the conclusion is robust.

1) In lines 282-284, although it is known that CD11b can bind to ICAM-1 on endothelial cells, the data from Figure 1 is not sufficient to support the claim, ideally a rescue assay should be performed, or test the binding ability of CD11b-/- neutrophils with tumor cells that overexpressed ICAM-1 compared to control tumor cells.

2) In line 288, the authors claimed to test metastasis by doing tail vein injection of tumor cells, but as authors may be aware, it only represents the post-intravasation part of the metastasis, the wording in line 288 needs to be more precise. 

Round 2

Reviewer 1 Report

Authors have responded fully to my comments.

Author Response

Thanks!

Reviewer 2 Report

The authors proposed that the involvement of the ICAM-1-uPAR-CD11b axis in tumor cell-neutrophil binding and subsequent metastasis formation of triple-negative breast cancer. However, the data are not sufficient to support their assumption

Major comments

#1. Figures 1 and 2. The authors demonstrated the data arising from using a single murine breast cancer cell line. As far as I know, this journal’s principle is to demonstrate the generality of the observations by using two distinct cell lines in a mouse experimental model.  Therefore, they should use additional cancer cell lines to prove the generality of their observations.

#4. Figure 2. The authors claimed the involvement of CD11b expressed in neutrophils by using CD11b knockout mice because neutrophil-specific CD11b knockout mice are not available at present. Instead of neutrophil-specific CD11b knockout mice, the authors should conduct similar experiments by using bone marrow chimeric mice between CD11b knockout and wild-type mice, in order to prove the involvement of CD11b expressed in bone marrow-derived cells.

Minor comments

#1. Figure 4B. shICAM-1 treatment decreased the amount of secreted uPAR only marginally. It is very doubtful on the biological significance of the observations. Moreover, the authors should clarify the molecular mechanisms underlying ICAM-1-mediated secretion of uPAR.

#2. Figure 5. The authors should examine whether similar observations are obtained by using human system.

Round 3

Reviewer 2 Report

The authors refused to modify the manuscript in response to the major comments on the previous version.